# Two mouse models carrying truncating mutations in *Magel2* show distinct phenotypes

**Daisuke Ieda[1], Yutaka Negishi[1], Tomomi Miyamoto[2], Yoshikazu Johmura[3], Natsuko Kumamoto[4], Kohji Kato[1,5], Ichiro Miyoshi[2¤], Makoto Nakanishi[3], Shinya Ugawa[4], Hisashi Oishi[2], Shinji Saitoh[1]***

**1** Department of Pediatrics and Neonatology, Nagoya City University Graduate School of Medical Sciences, Nagoya, Japan, **2** Department of Comparative and Experimental Medicine, Nagoya City University Graduate School of Medical Sciences, Nagoya, Japan, **3** Division of Cancer Cell Biology, Department of Cancer Biology, Institute of Medical Science, The University of Tokyo, Tokyo, Japan, **4** Department of Anatomy and Neuroscience, Nagoya City University Graduate School of Medical Sciences, Nagoya, Japan, **5** Department of Pediatrics, Nagoya University Graduate School of Medicine, Nagoya Japan

¤ Current address: Department of Laboratory Animal Medicine, Tohoku University Graduate School of Medicine, Sendai, Japan

* ss11@med.nagoya-cu.ac.jp

**Data Availability Statement:** All relevant data are within the manuscript and its Supporting Information files.

## Abstract

Schaaf-Yang syndrome (SYS) is a neurodevelopmental disorder caused by truncating variants in the paternal allele of *MAGEL2*, located in the Prader-Willi critical region, 15q11-q13. Although the phenotypes of SYS overlap those of Prader-Willi syndrome (PWS), including neonatal hypotonia, feeding problems, and developmental delay/intellectual disability, SYS patients show autism spectrum disorder and joint contractures, which are atypical phenotypes for PWS. Therefore, we hypothesized that the truncated *Magel2* protein could potentially produce gain-of-function toxic effects. To test the hypothesis, we generated two engineered mouse models; one, an overexpression model that expressed the N-terminal region of *Magel2* that was FLAG tagged with a strong ubiquitous promoter, and another, a genome-edited model that carried a truncating variant in *Magel2* generated using the CRISPR/Cas9 system. In the overexpression model, all transgenic mice died in the fetal or neonatal period indicating embryonic or neonatal lethality of the transgene. Therefore, overexpression of the truncated *Magel2* could show toxic effects. In the genome-edited model, we generated a mouse model carrying a frameshift variant (c.1690_1924del; p(Glu564Serfs*130)) in *Magel2*. Model mice carrying the frameshift variant in the paternal or maternal allele of *Magel2* were termed *Magel2*[P:fs] and *Magel2*[M:fs], respectively. The imprinted expression and spatial distribution of truncating *Magel2* transcripts in the brain were maintained. Although neonatal *Magel2*[P:fs] mice were lighter than wildtype littermates, *Magel2*[P:fs] males and females weighed the same as their wildtype littermates by eight and four weeks of age, respectively. Collectively, the overexpression mouse model may recapitulate fetal or neonatal death, which are the severest phenotypes for SYS. In contrast, the genome-edited mouse model maintains genomic imprinting and distribution of truncated *Magel2* transcripts in the brain, but only partially recapitulates SYS phenotypes. Therefore, our results imply that simple gain-of-

**Funding:** This study was partly supported by JSPS KAKENHI Grant-in-Aid for Early-Career Scientists (JP18K15682). There was no additional external funding received for this study.

**Competing interests:** The authors have declared that no competing interests exist.

function toxic effects may not explain the patho-mechanism of SYS, but rather suggest a range of effects due to *Magel2* variants as in human SYS patients.

## Introduction

In 2013, the first four individuals with truncating variants in the paternal allele of *MAGEL2* were reported, and later described as having Schaaf-Yang syndrome (SYS, OMIM#615547). The phenotypes of SYS patients overlap those of Prader-Willi syndrome (PWS, OMIM#176270), including neonatal hypotonia, feeding problems and developmental delay/intellectual disability (DD/ID) [1]. Additionally, SYS patients show autism spectrum disorder (ASD) and contractures of the small finger joints, which are atypical phenotypes for PWS [2].

PWS occurs as the result of absence of expression of paternal genes from chromosome 15q11.2-q13 [3, 4]. Chromosome 15q11.2-q13 contains paternal-only expressed genes encoding polypeptides (*MKRN3*, *MAGEL2*, *NDN*, *NPAP1* and *SNURF-SNRPN*) [1]. It also contains snoRNAs (*SNORD115*, *116*) which show paternal-only expression [5]. Recently, it was revealed that a paternal deletion of *SNORD116* is responsible for PWS [6]. *MAGEL2* is not expressed in patients with PWS. Therefore, a loss-of-function in *MAGEL2* should be associated with PWS. Nevertheless, SYS patients generally show more severe phenotypes than typical PWS patients. Additionally, patients with a paternally inherited deletion including *MAGEL2*, but not *SNRPN/SNORD116*, have a milder phenotype than those with truncating variants in *MAGEL2* [7, 8]. Thus, a gain-of-function mechanism in *MAGEL2* was suggested as the pathological mechanism underlying SYS [9].

Human *MAGEL2*, and its mouse ortholog *Magel2*, are GC-rich, single-exon, maternally imprinted genes that are exclusively expressed from the unmethylated paternal allele. *MAGEL2* and *Magel2* encode putative proteins of 1249 and 1284 amino acids, respectively, which are highly homologous (Fig 1) [9, 10]. In SYS, more than half of the patients carry a truncating variant in nucleotides c.1990-1996, which is upstream of the region encoding the C-terminus of the proline-rich region in *MAGEL2*. Notably, c.1996dupC is the most common and severe variant in SYS patients [9]. In mice, *Magel2* RNA is expressed at low levels throughout the brain, but shows the highest expression in hypothalamic regions, especially the paraventricular nucleus (PVN) and suprachiasmatic nucleus (SCN). A mouse model has been generated by inactivating *Magel2* in C57BL/6 mice with the use of a lacZ knock-in allele with paternal inheritance [11, 12]. *Magel2*-null mice have reduced embryonic viability, but otherwise normal embryonic growth in survivors, followed by postnatal growth retardation. In their later development, they even showed more weight gain compared to littermates [11]. Such mild phenotypes in *Magel2*-null mice did not recapitulate those of SYS, but may represent those of patients with a deletion of the entire *MAGEL2* gene.

Therefore, we generated mouse models to test the hypothesis that the truncating *MAGEL2* protein could potentially produce gain-of-function toxic effects. Assuming that mice carrying a truncating variant in *Magel2* have a more severe phenotype than *Magel2*-null mice, we generated two types of mouse models: a transgenic mouse that overexpressed the N-terminal region of *Magel2*, and a genome-edited mouse expressing truncating *Magel2* under the intrinsic promoter.

## Materials and methods

### Vector construction

**pCAGGS1-*Magel2*-FLAG.** We generated an overexpression model that overexpressed the N-terminal region of *Magel2* (amino acid residues 1–437). We amplified a 1311bp fragment

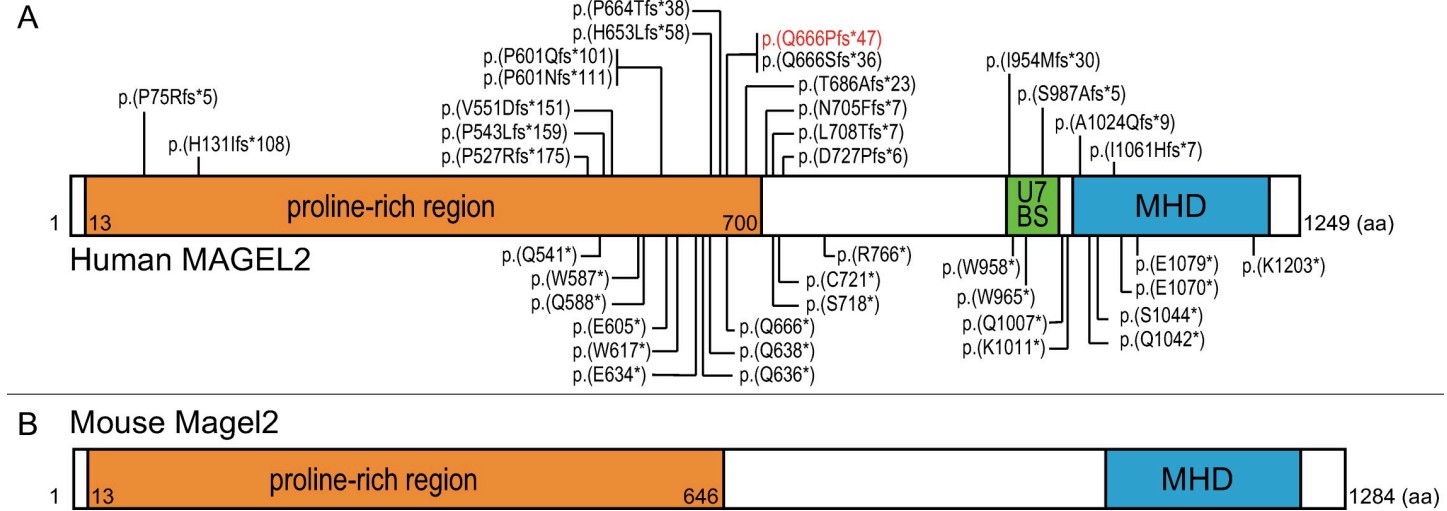

**Fig 1. Schematic structure of the human *MAGEL2* and mice *Magel2*.** (A) Human *MAGEL2* contains a proline-rich region (residues 13–700), USP7 binding site (U7BS: residues 949–1004), and MAGE homolog domain (MHD: residues 1020–1219). Truncating variants reported previously are indicated by their positions (top; frameshift variants, bottom: nonsense variants). The mutation hotspot is located at nucleotides c.1990-1996. Over half of SYS patients carried c.1996dupC:p.(Q666Pfs*47) in *MAGEL2* (in red text). (B) Mouse *Magel2* contains proline-rich region (residues 13–646) and MHD (residues 1052–1251).

encoding the N-terminal region of *Magel2* with a FLAG tag at the C-terminus by polymerase chain reaction (PCR). PCR was performed with mouse genomic DNA, AmpliTaq Gold 360 Master Mix (Thermo Fisher Scientific, Waltham, MA), and primers F1 and R1. Primers F1 and R1 contained recognition sites for *Eco*RI and *Xho*I, respectively. As an antibody specific to *Magel2* protein was not available, we inserted a FLAG-tag at the C-terminus of the truncated *Magel2* (S1 Fig). To express truncated *Magel2* under the control of the CAG promoter, the product was subcloned into *pCAGGS1* containing a modified chicken actin promoter with the CAG promoter, kindly provided by Dr. J. Miyazaki (Osaka University), using *Eco*RI and *Xho*I cloning sites (Fig 2A). The construct was named *pCAGGS1-Magel2-FLAG*. The construct was linearized by digestion with *Sal*I and *Hind*III prior to microinjection into fertilized eggs.

**pX330-*Magel2*.** We generated a genome-edited model that carried a truncating variant in intrinsic *Magel2*. We selected 5´-CCACAGGAGCTCCCGGTGCCACA-3´ as a target sequence for CRISPR/Cas9 which located on c.1702-1724, c.1720-1742, c.1810-1832, c.1882-1904 and

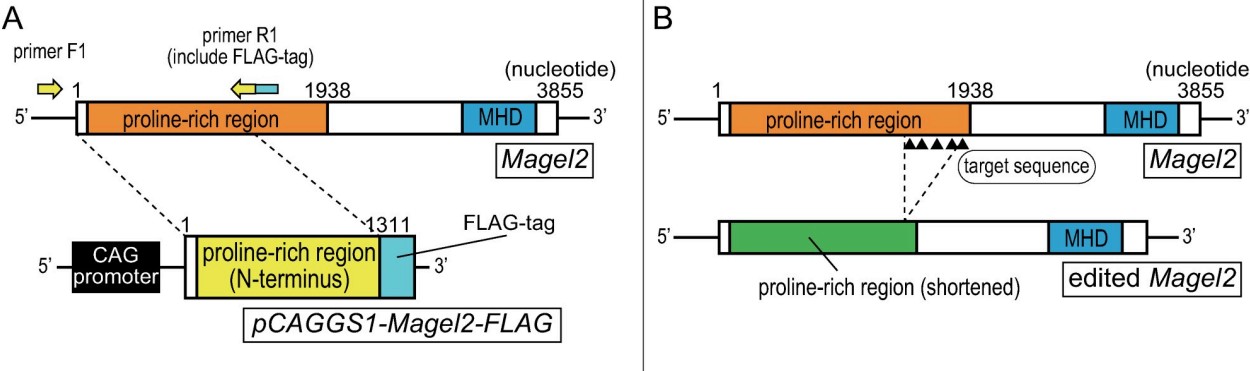

**Fig 2. Strategies to generate transgenic mice.** (A) Strategy to generate the overexpression model which overexpresses the N-terminal region of *Magel2* with FLAG-tag. (B) Strategy to generate genome-edited model which carries truncating variant in *Magel2*. Target sequence for CRISPR/Cas9 (5´-CCACAGGAGCTCCCGGTGCCACA-3´) is located on c.1702-1724, c.1720-1742, c.1810-1832, c.1882-1904 and c.1900-1922 (black triangles).

c.1900-1922 in *Magel2*. Target sequences were located near the C-terminus of the proline-rich region in *Magel2*. The *pX330* plasmid (Addgene plasmid #42230) carries both guide RNA and Cas9 expression unit. *Magel2*-CRISPR-F (5´-caccTGTGGCACCGGGAGCTCCTG-3´) and *Magel2*-CRISPR-R (5´-aaacCAGGAGCTCCCGGTGCCACA-3´) oligo DNAs were annealed and subcloned into *pX330* with *Bbs*I cloning site as described previously [13]. The plasmid was designated as *pX330-Magel2*.

**p2color-*Magel2*.**    The *p2color* vector (RDB13948, RIKEN BRC, Tsukuba Japan) contains a multiple cloning site target site between RFP- and GFP-encoding DNA sequences. *Magel2*-screening-F (aattTGTGGCACCGGGAGCTCCTGTGGCACCGG) and *Magel2*-screening-R (ggccCCGGTGCCACAGGAGCTCCCGGTGCCACA) oligo DNAs were annealed and subcloned into the *p2color vector* with *Eco*RI/*Not*I cloning sites as described previously [13]. The plasmid was designated as *p2color-Magel2*. *p2color-Magel2* was used for the RFP-GFP reporter assay which confirmed the cleavage activity of *pX330-Magel2*.

### Western blotting

We transfected *pCAGGS1-Magel2-FLAG* into HEK293 cells using Lipofectamine 2000 (Thermo Fisher Scientific), and confirmed its expression by western blot analysis as previously described using primary antibodies against FLAG (diluted 1:1000; Cell Signaling Technology, Danvers, MA) and GAPDH (diluted 1:10,000; Cell Signaling Technology), and a horseradish peroxidase–conjugated secondary antibody (GE Healthcare, Little Chalfont, UK) [14].

### RFP-GFP reporter assay

We co-transfected HEK293T cells with *pX330-Magel2* and *p2color-Magel2* using Lipofectamine 2000 following the manufacturer's protocol. In this assay, the GFP sequence is fused to the target site out of frame, and functional GFP is expressed only when CRISPR/Cas9 induces a double-strand break at the target site, whose repair by error-prone non-homologous end joining gives rise to indels that often result in a frameshift variant (S2 Fig) [15].

### Mouse breeding and handling

C57BL/6N mice and ICR mice were purchased from Japan SLC, Inc. (Hamamatsu, Japan). Mice were kept in plastic cages under pathogen-free conditions in a room maintained at 23 ± 2°C under 12:12 light dark conditions. Mice were weaned at four weeks of age then housed at 1–7 mice per cage with food (Oriental Bio Service, Kyoto, Japan) and filtered water *ad libitum*. Mice used for weight measurement were housed at 4–5 mice per cage after weaning. Mice were euthanized at the appropriate time points with carbon dioxide followed by cervical dislocation. All experimental procedures conformed to the Regulations for Animal Experimentation at Nagoya City University, reviewed by the Institutional Laboratory Animal Care and Use Committee of Nagoya City University, and approved by the provost (Protocol Number: H27M-14, H28M-70, H29M-64).

### Microinjection

Four-week-old C57BL/6N females were superovulated with 7.5 IU of pregnant mare serum gonadotropin and human chorionic gonadotropin and mated with 10-week-old C57BL/6N males. Pronuclear-stage eggs were injected with the linearized transgene, cultivated in KSOM overnight, and then transferred into the oviducts of 7-week-old pseudopregnant ICR females. On the 19[th] day of pregnancy, we sacrificed ICR females and performed cesarean section. Thus, we obtained live-born pups and stunted embryos.

## Genomic DNA analysis

Genomic DNA was extracted from mouse tails or embryos with KAPA Mouse Genotyping Kit (Nippon Genetics Co., Ltd., Tokyo, Japan) according to manufacturer's protocol. Mice were anesthetized with 1% isoflurane, and tail tips taken at four weeks of age. PCR on genomic DNA was performed with AmpliTaq Gold 360 Master Mix and primer pairs. The PCR products were separated using 3% agarose gel electrophoresis. The primers F2 (5´-CAGTATCAGG AGCACCAA-3´) and R2 (5´-ATCCTTGTAGTCCATAGGAC-3´) were used for the overexpression model (S3 Fig). As primer R2 was designed within FLAG-tag sequence, DNA from mice carrying the transgene were specifically amplified by PCR. The primers F3 (5´-CCAA CTGTCTATCCCAAT-3´) and R3 (5´-TGCCAGAAGTGAGGAGGT-3´) were used for the genome-edited model (S4 Fig). In mice carrying indels in the C-terminus of the proline-rich region, the length of amplified DNA was shorter than those of wildtype. PCR products were sequenced with BigDye Terminator v3.1 Cycle Sequencing Kit and SeqStudio Genetic Analyzer (Thermo Fisher Scientific). Mice carrying the transgene were used for subsequent mating.

## Reverse Transcription-PCR (RT-PCR) analysis

Total RNA was isolated from hypothalamus of neonatal mice (P10) using RNeasy Plus Mini Kit (Qiagen, Hilden, Germany). We removed genomic DNA from total RNA products by using recombinant DNase I (Takara Bio Inc., Shiga, Japan). Reverse transcription was performed using purified RNA and SuperScript IV Reverse Transcriptase (Invitrogen, Carlsbad, CA). All processes were performed following the manufacturer's protocol. After reverse transcription, complementary DNA (cDNA) was amplified with primer F3 and R3, and AmpliTaq Gold 360 Master Mix. The RT-PCR products were separated using 3% agarose gel electrophoresis.

## *In situ* hybridization (ISH)

ISH of *Magel2* mRNA was performed on the brains of young-adult (8–9 weeks) male mice. DNA template for probe (targeted at bases c.1059-1679 of the *Magel2* mRNA) was amplified from C57BL/6N mice DNA, with *Magel2*-probe primer pairs (5´-TGTACCACAAGCCCC CCA-3´ and 5´-GGGGCCTGGCCTTTGG-3´), and AmpliTaq Gold 360 Master Mix. The DNA template was subcloned into pGEM-T easy vector (Promega, Madison, WI, USA) following the manufacturer's protocol (S5 Fig). We synthesized [$^{35}$S]UTP-labeled *Magel2* antisense strand probes with T7 RNA polymerase after DNA construct linearization by digestion with *Sal*I. We also synthesized $^{35}$S-labeled *Magel2* sense strand probes with SP6 RNA polymerase after DNA construct linearization by digestion with *Nco*I. The antisense strand and the sense strand were used for the cRNA probe and negative control, respectively. Cryosections (20 μm thick) were cut from freshly frozen mouse brains, and fixed in 4% formaldehyde in phosphate buffer (0.1 M, pH 7.4) with proteinase K (10 μg/mL). The sections were acetylated with acetic anhydride, dehydrated in ascending alcohol series, and air-dried. They were incubated in hybridization buffer (50% formamide, 0.3 M NaCl, 20 mM Tris-HCl, 10% dextran sulfate, Denhardt's solution, 500 μg/mL yeast tRNA, 20 mM dithiothreitol, and 200 μg/mL salmon testis DNA) with the synthesized RNA probes for 12 hours at 55˚C. After hybridization, they were washed with 50% formamide/2× standard sodium citrate (SSC) at 65˚C and incubated with 1 μg/mL RNase A in RNase buffer (0.5 M NaCl, 10 mM Tris–HCl, and 1 mM EDTA, pH 8.0) for 30 min at 37˚C. Subsequently, they were washed in 50% formamide/2× SSC at 65˚C, rinsed with 2X SSC and 0.1X SSC, dehydrated in alcohol, and air-dried [16]. The slides were then stained with hematoxylin and eosin, and images captured with a CCD camera

**Table 1. Genotype distribution of live-born offspring and stunted embryos in the *Magel2* overexpression model.**

|  | Transgene-positive | Transgene-negative | Total |
|---|---|---|---|
| Live-born pups | 3 (6.1%) | 49 | 52 |
| Stunted embryos | 22 (75.9%) | 7 | 29 |

(OLYMPUS, Tokyo, Japan) connected to a stereomicroscope (Carl Zeiss, Oberkochen, Germany).

## Results

### Generation of the *Magel2* overexpression model

First, we transfected *pCAGGS1-Magel2-FLAG* into HEK293T cell and confirmed its expression by western blot analysis (S6 Fig). Second, *pCAGGS1-Magel2-FLAG* vector was injected into the pronuclei of fertilized oocytes to obtain mice that overexpressed the N-terminal region of *Magel2* (amino acid residues 1–437) with a FLAG tag. We obtained 52 live-born pups and 29 stunted embryos from six litters. Although three of the 52 pups (5.8%) carried the transgene, two died immediately after birth and one exhibited a small body size and poor suck, and died at P13. None of the surviving 49 pups carried the transgene. Furthermore, 22 of 29 stunted embryos (75.9%) carried the transgene (Table 1). The probability of being transgene-positive was statistically significant between live-born pups and stunted embryos ($P < 0.001$, Pearson's chi-squared test).

### Generation of the *Magel2* genome-edited model

*pX330-Magel2* was injected into the pronuclei of fertilized oocytes to obtain mice carrying frameshift variants in *Magel2* target sites. From six litters, we obtained 24 pups, of which 20 survived and were genotyped. There were 12 unique *Magel2* variants across the 20 pups (Fig 3A). We selected a male mouse carrying a homozygous frameshift variant in *Magel2* (c.1690_1924del;p(Glu564Serfs*130)) as the founder mouse (Fig 3B). We then mated the founder mouse with wildtype females, and obtained mice carrying heterozygous frameshift variant in *Magel2*. Next, we mated affected males with wildtype females and obtained model mice carrying a paternal frameshift variant in *Magel2*, which were termed '*Magel2*^P:fs^'. We also mated wildtype males with affected females and obtained control mice carrying a maternal frameshift variant in *Magel2*, which were termed '*Magel2*^M:fs^'. Littermates without a variant in *Magel2* were termed '*Magel2*^+^' (Fig 3C). In *Magel2*^P:fs^ mice, there were no obvious abnormality in physical findings, including contracture, which is a distinctive phenotype of human SYS patients.

### Birth rate of the genome-edited mouse model

To investigate the birth rate of *Magel2*^P:fs^ and *Magel2*^M:fs^ mice we mated affected males with wildtype females, and obtained 201 live-born pups from 27 litters. Eighty-five pups (42.3%) were *Magel2*^P:fs^. We also mated wildtype males with affected females and obtained 43 live-born pups from 6 litters. Twenty-two pups (51.2%) were *Magel2*^M:fs^. The birth rate of *Magel2*^P:fs^ was less than expected, but there was no significant difference between *Magel2*^P:fs^ and *Magel2*^M:fs^ ($P = 0.287$, power 0.186, Pearson's chi-squared test; Table 2).

### Expression of mRNA in *Magel2*

We performed RT-PCR on mRNA from newborn mouse brains. *Magel2*^P:fs^ and *Magel2*^M:fs^ mice expressed truncating *Magel2* mRNA and normal *Magel2* mRNA, respectively (Fig 4A).

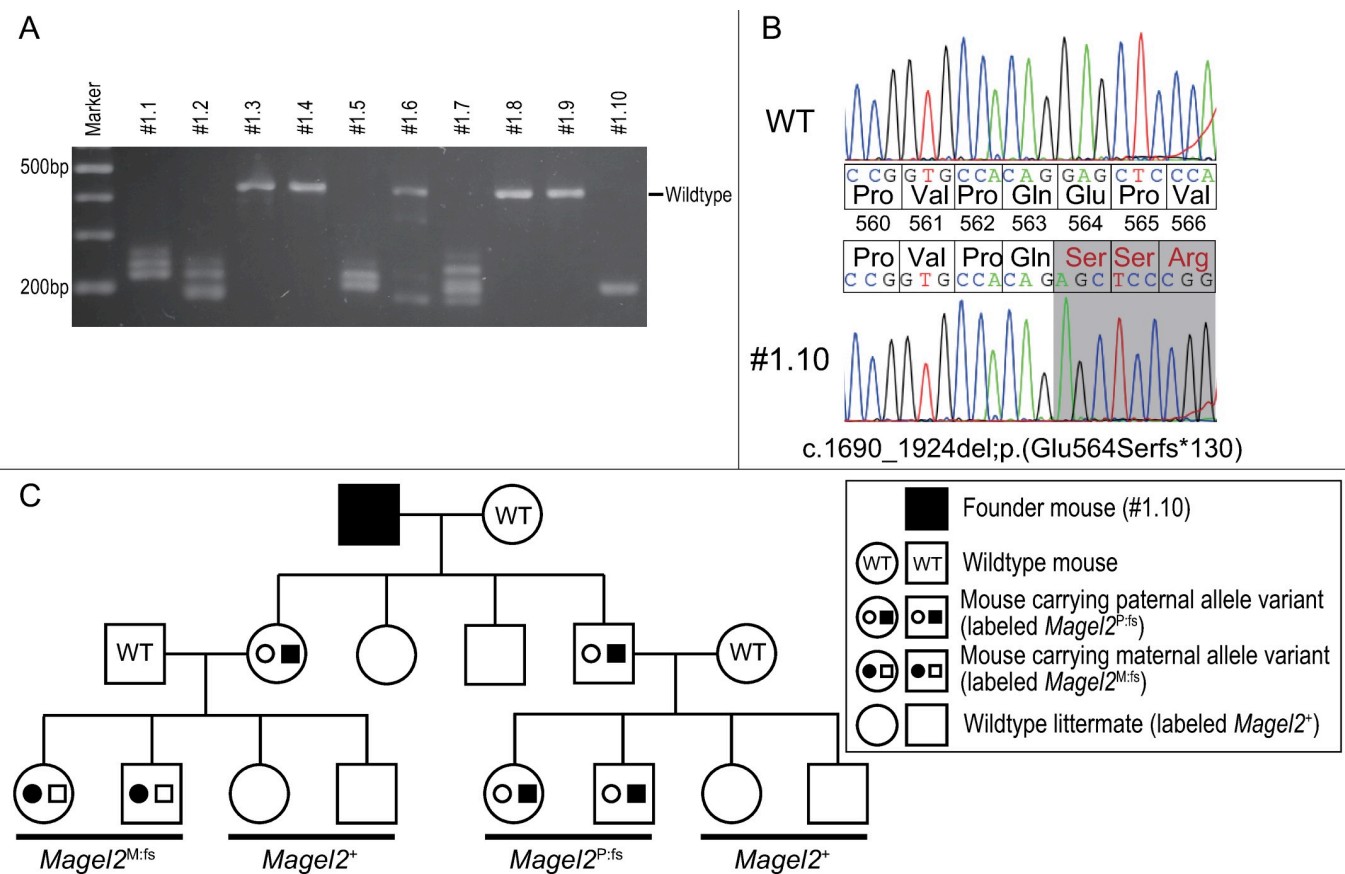

**Fig 3. Generation of a mouse model carrying a frameshift variant in *Magel2*.** (A) We obtained genome-edited mice carrying different variants in *Magel2*. We selected #1.10 as the founder mouse. (B) Comparison of the base sequence and amino acid residues in *Magel2*. Founder mouse #1.10 carried a homozygous frameshift variant in *Magel2*. (C) The pedigree of our mouse model. Mice carrying a variant in the paternal allele of *Magel2* were termed '*Magel2*^P:fs'. Mice carrying a variant in the maternal allele of *Magel* were termed '*Magel2* ^M:fs'. Littermates not carrying a variant in *Magel2* were termed '*Magel2*^+'.

Thus, the paternal allele of *Magel2* was expressed and the maternal allele of *Magel2* was silenced.

## Localization of *Magel2* mRNA in the young-adult mouse brain

We performed ISH on the brains of *Magel2*^P:fs and wildtype young-adult mice. *Magel2* mRNA was detected in the SCN and PVN of the hypothalamus in both groups (Fig 4B, S7 Fig). Thus, *Magel2*^P:fs mice did not have an altered localization of *Magel2* mRNA.

## Body weight in the genome-edited mouse model

In neonates (P10), *Magel2*^P:fs pups were lighter than *Magel2*^+ (5.44 ± 0.12 g vs 6.11 ± 0.13 g, $P = 0.0003$, Welch's *t*-test). By contrast, there was no difference between *Magel2*^M:fs and

**Table 2. Genotype distribution of offspring born in the genome-edited model.**

|  | Variant-positive | Variant-negative | Total |
|---|---|---|---|
| *Magel2*^P:fs | 85 (42.3%) | 116 | 201 |
| *Magel2*^M:fs | 22 (51.2%) | 21 | 43 |

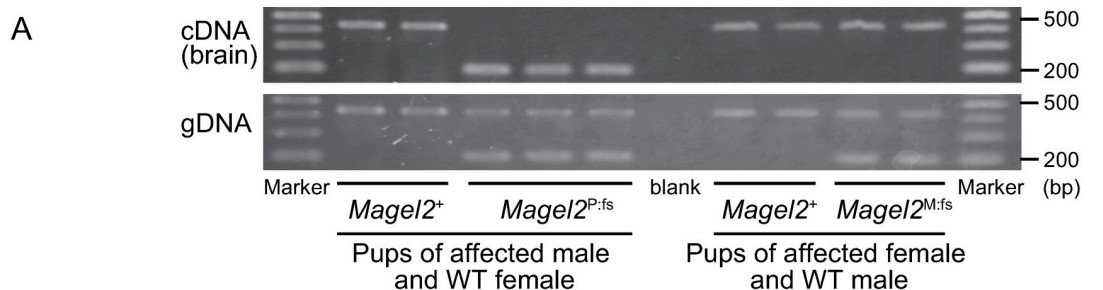

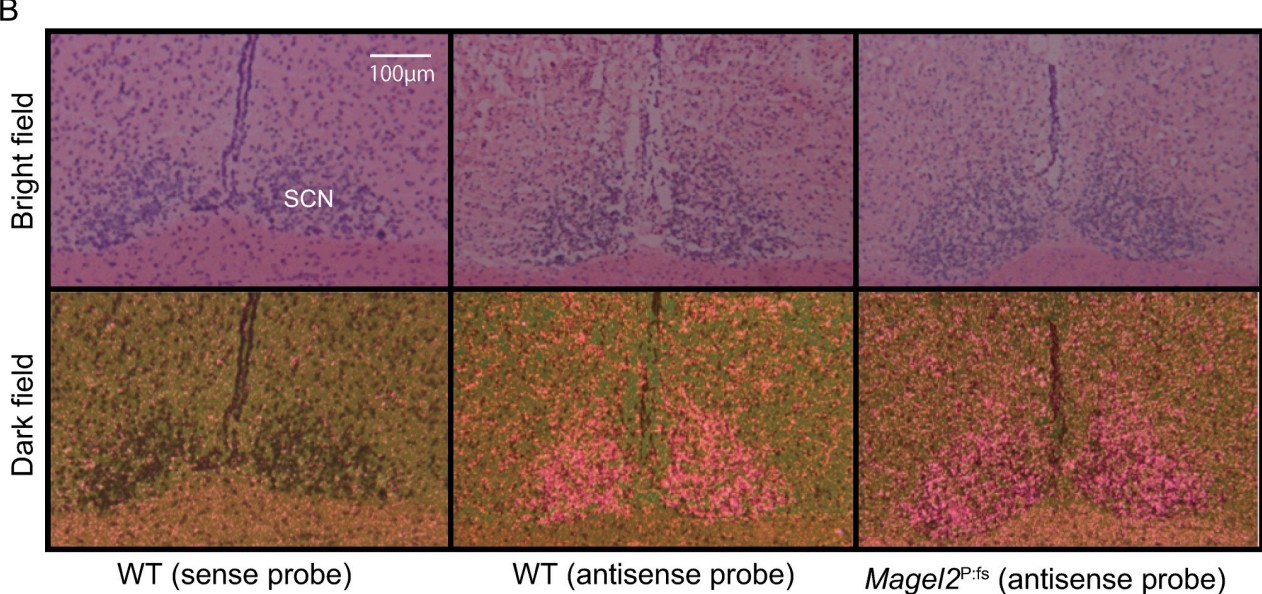

**Fig 4. Expression and distribution of *Magel2* in the mouse brain.** (A) Expression of *Magel2* transcripts in the neonatal mouse brain. Only the paternal allele of *Magel2* is expressed in the brain. (B) Distribution of *Magel2* transcripts in young-adult brains was similar in WT and *Magel2*^P:fs mice. *Magel2* mRNAs were expressed in the SCN of the hypothalamus in both groups. SCN: suprachiasmatic nucleus. Scale bar: 100 μm.

*Magel2*+ pups (6.00 ± 0.13 g vs 5.72 ± 0.11 g, $P = 0.058$, Welch's *t*-test; Fig 5A). For males, *Magel2*^P:fs mice were lighter than their *Magel2*+ male littermates at four weeks of age (13.71 ± 0.49g vs 15.84 ± 0.50g, $P = 0.0032$, Welch's *t*-test). By eight weeks of age however, the weight of *Magel2*^P:fs males was similar to *Magel2*+ males (Fig 5B). For females, there was no difference between *Magel2*^M:fs mice and *Magel2*+ female littermates at four weeks of age (13.84 ± 0.34 g vs 13.07 ± 0.31 g, $P = 0.38$, Welch's *t*-test; Fig 5C).

## Discussion

*MAGEL2* is located in the PWS critical region and typically deleted in PWS patients. Therefore, loss-of-function phenotypes with *MAGEL2* are likely to be included in typical PWS patients. However, most SYS patients with a truncating variant in *MAGEL2* show more severe clinical features than PWS patients. Indeed, our previous study identified six patients with SYS with a truncating variant, including the common c.1996dupC, and all six patients showed severe intellectual disability and complication of joint contracture, which are atypical for PWS [17]. Thus, we hypothesized that the truncated Magel2 protein could potentially produce gain-of-function toxic effects, and we generated two types of mouse models expressing truncating mutations of *Magel2*.

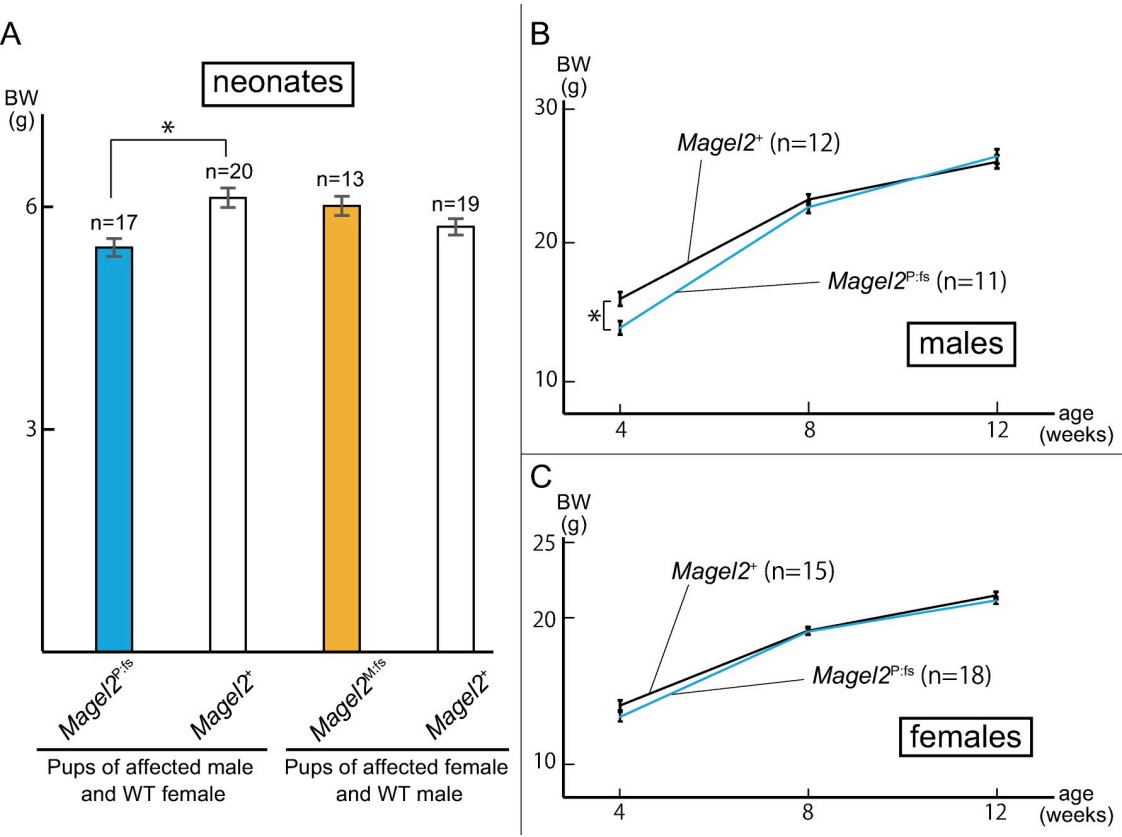

**Fig 5. Comparison of body weight of affected and WT mice.** (A) Mean weight ± SEM at P10. The weight of *Magel2*P:fs was reduced compared to *Magel2*+ (5.44 ± 0.12 g vs 6.11 ± 0.13 g, *P* = 0.0003). There was no difference between *Magel2*M:fs and *Magel2*+ mice (6.00 ± 0.13 g vs 5.72 ± 0.11 g, *P* = 0.058). Both sexes were included. (B, C) Mean weight ± SEM at four, eight and 12 weeks of age in both sexes. At four weeks, *Magel2*P:fs males were lighter than *Magel2*+ (13.71 ± 0.49 g vs 15.84 ± 0.50 g, *P* = 0.0032), but by eight weeks of age, there was no difference. In females, there was no difference between *Magel2*P:fs mice and *Magel2*+ at four, eight and 12 weeks of age.

First, we generated an overexpression model which expressed truncating *Magel2* under the control of the CAG promoter, and found that all transgenic mice died in fetal or neonatal period. In normal mouse embryos, *Magel2* is expressed in the hypothalamus, cerebral cortex, and spinal cord [18]. In contrast, our model mice expressed truncated *Magel2* under the CAG promoter, which induced much higher expression levels ubiquitously. Therefore, we assumed that the expression of truncated *Magel2* in various types of organs were responsible for fetal death. We tried to extract protein from the brain of dead mice in the overexpression model, and to detect truncated Magel2 protein with FLAG tag. However, we were not able to detect FLAG tag signal by western blot analysis as the extracted proteins were denatured. Therefore, we were not able to assess the patho-mechanism of fetal or neonatal death in the overexpression mouse model. Nevertheless, the overexpression model revealed the toxic effects of overexpression of truncated *Magel2* in the fetal period. It is intriguing that Mejlachowics et al. reported that the c.1996delC variant in *MAGEL2* was responsible for fetal death at 24 to 27 weeks of gestation in human [19], indicating the severe toxic effects of the specific truncated *MAGEL2*. Our overexpression model may, at least in some degree, recapitulate the most severe of the toxic effects of truncated *MAGEL2*.

Next, we generated a genome-edited mouse model carrying a frameshift variant in *Magel2* (c.1690_1924del;p(Glu564Serfs*130)) with the CRISPR/Cas9 system. As the paternal allele of

*Magel2* is expressed, but the maternal allele is silenced, we classified the model mice into those carrying a paternal allele variant (*Magel2*[P:fs]) and those carrying a maternal allele variant (*Magel2*[M:fs]).

Kozlov et al. generated a mouse model with inactivate paternal *Magel2* with the use of a lacZ knock-in allele, and *Magel2* knockout model displayed 10% postnatal lethality [12]. We investigated the birth rate of *Magel2*[P:fs] and *Magel2*[M:fs], but there was no difference between the two groups although statistical power was not high.

Imprinting regions, including *Magel2*, are governed by imprinting centers that regulate parent-of-origin epigenotypes and gene expression patterns [20]. Matarazo et al. reported the loss of imprinting and the expression of the maternal allele of *Magel2* in a mouse model with a deletion of the paternal allele of *Magel2*, including its promoter [21]. We confirmed genome imprinting by RT-PCT on newborn mouse brain mRNA, and revealed that *Magel2*[P:fs] expressed truncated *Magel2*, and *Magel2*[M:fs] expressed full-length *Magel2*. That suggested that our mouse model maintained the genome imprinting mechanism of *Magel2*.

*Magel2* expression is specifically localized to the SCN and PVN in the hypothalamus [12, 18]. We performed ISH to compare the distribution of *Magel2* mRNA in wildtype and *Magel2*[P:fs] males. There was no difference in the distribution of *Magel2* mRNA. Those results suggested that our model mice maintained the distribution of *Magel2* mRNA in the brain. As we do not have a specific anti-Magel2 antibody, we could not assess the expression of the Magel2 protein.

We measured the body weight of genome-edited model mice compared with their littermates (*Magel2*[+]). At P10, *Magel2*[P:fs] pups were statistically lighter than *Magel2*[+]. Although *Magel2*[P:fs] males were statistically lighter than *Magel2*[+] males at four weeks of age, their body weight caught up with those of *Magel2*[+] males at eight weeks of age. Furthermore, the body weight of *Magel2*[P:fs] females was similar to that of *Magel2*[+] females at four weeks of age. Bischof et al. reported that *Magel2*-null mice exhibited neonatal growth retardation and excessive weight gain after weaning, and their growth abnormality was similar to PWS [11]. In contrast, it was reported that 97% of SYS patients exhibited poor suck in infancy, but only 22–41% exhibited excessive weight gain [9, 22]. Our mouse model showed growth retardation in neonates, but they did not show excessive weight gain after weaning. They may partially recapitulate human SYS phenotype in terms of characteristic growth abnormality.

The genome-edited mouse model did not show obvious abnormality in physical findings. In humans, genotype-phenotype association in *MAGEL2* has been discussed previously. McCarthy et al. mentioned that c.1996dupC in *MAGEL2* is the most common variant in SYS. Patients carrying c.1996dupC in *MAGEL2* showed a higher prevalence of joint contractures, feeding difficulties, and severe ID/DD than patients carrying other variants in *MAGEL2*. They mentioned that the severity of SYS depended on the specific location of the truncating mutation. *MAGEL2* and *Magel2* are single exon genes, and mutations leading to a premature stop codon are predicted not to cause nonsense-mediated mRNA decay. The pathogenic effect of the truncated *MAGEL2* protein may differ depending on the precise location of the mutation in *MAGEL2* [9]. Our mouse model carrying the c.1690_1924del variant in paternal *Magel2* may produce truncated *Magel2* protein which shows a milder toxic effect.

Our genome-edited mouse model showed almost comparable or less severe phenotypes to previously reported *Magel2* null mice, and failed to recapitulate the common phenotype of SYS. This may be due to the position of the variant we made and the wide clinical spectrum of human SYS patients. Thus, the gain-of-function hypothesis remains unsolved. Nevertheless, in our genome-edited mouse model, we showed the maintenance of imprinted expression and the distribution of the truncated *Magel2* transcripts in the mouse brain.

There are several limitations of the study. First, although we analyzed *Magel2* mRNA, we did not analyze Magel2 protein due to the lack of a specific anti-Magel2 antibody. Second, we were not able to investigate the patho-mechanism of fatal or neonatal death of the overexpression model because we were unable to obtain purified brain proteins. Third, the analyses of our mouse models are not exhaustive. It is known that human SYS patients have ASD and arthrogryposis, and Magel2-null mice have altered circadian rhythm, reduced motor activity, and increased adiposity [2, 12, 23]. Therefore, behavioral, anatomical and serological tests are required for our mouse model in the future.

## Conclusion

We generated two types of mouse models carrying a truncating variant in *Magel2*. The overexpression model was embryonic or neonatal lethal, indicating toxic effects of overexpression of the truncated *Magel2*. The genome-edited model maintained genomic imprinting and distribution of truncated *Magel2* transcripts in the brain, and only partially recapitulate SYS phenotypes. Our results suggest that not simple gain-of-function toxic effects, but rather varied effects due to the position and type of *MAGEL2* variants, underlie the patho-mechanism of SYS.

## Supporting information

**S1 Fig. Scheme for incorporating a FLAG-tag into the N-terminal region of Magel2, including primer details.**
(EPS)

**S2 Fig. Reporter system for confirming CRISPR activity.** (A) Scheme of the RFP-GFP reporter-based assay for measuring the activity of the CRISPR/Cas9 system. The CRISPR/Cas9 system induces double-strand breaks for target sequence, with frameshift variants incorporated after non-homologous end joining. (B) Robust EGFP signals were only seen in HEK293T cells co-transfected with both *p2color-Magel2* and *pX330-Magel2*.
(EPS)

**S3 Fig. Genotyping scheme for the overexpression model.** Primer R2 contains the sequence complementary to the terminus of truncated *Magel2* and FLAG-tag sequence. DNA from transgenic mice is specifically amplified with primers F2 and R2.
(EPS)

**S4 Fig. Genotyping scheme for genome-edited model.** In wildtype DNA, PCR with primers F3 and R3 was predicted to produce an amplicon of 438bp. An amplicon would be predicted to be shorter than that of wildtype when *Magel2* was edited by CRISPR/Cas9 system.
(EPS)

**S5 Fig. DNA construct for RNA probe preparation.** *Magel2* DNA (c.1059-1679) was subcloned into pGEM-T easy vector. *Magel2* antisense strand for the cRNA probe and sense strand for negative control were synthesized by T7 and SP6 RNA polymerase, respectively.
(EPS)

**S6 Fig. Western blot analysis for HEK293 cells transfected with a construct overexpressing truncated *Magel2-FLAG* tag and untreated cells.**
(EPS)

**S7 Fig. Expression of *Magel2* in PVN by *in situ* hybridization.** Distributions of *Magel2* transcripts in PVN of wild-type mice and *Magel2*p.fs mice are shown in middle and right panels,

respectively. Left panels, sense controls. Scale bar: 100 μm.
(EPS)

**S1 Raw images.**
(PDF)

## Acknowledgments

We wish to thank all members of the laboratories of Department of Pediatrics and Neonatology, Nagoya City University Graduate School of Medical Sciences, for their assistance. We also acknowledge the assistance of the Research Equipment Sharing Center at the Nagoya City University.

## Author Contributions

**Conceptualization:** Shinji Saitoh.

**Investigation:** Daisuke Ieda, Yutaka Negishi, Tomomi Miyamoto, Yoshikazu Johmura, Natsuko Kumamoto, Kohji Kato, Ichiro Miyoshi, Makoto Nakanishi, Shinya Ugawa, Hisashi Oishi.

**Project administration:** Shinji Saitoh.

**Writing – original draft:** Daisuke Ieda, Shinji Saitoh.

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
