## [Decision Letter · Decision Letter 0]

19 May 2020

PONE-D-20-10964

Two mouse models carrying truncating mutations in Magel2 show distinct phenotypes

PLOS ONE

Dear Dr. Saitoh,

Thank you for submitting your manuscript to PLOS ONE. After careful consideration, we feel that it has merit but does not fully meet PLOS ONE’s publication criteria as it currently stands. Therefore, we invite you to submit a revised version of the manuscript that addresses the points raised during the review process.

I would like to congratulate you on your studying the hypothesis that mice carrying a truncating variant in Magel2 have a more severe phenotype than Magel2-null mice, with the generation of two types of mouse models.

This is most straight-forward.

Unfortunately, neither mouse model recapitulates the specific human phenotype associated with paternally-inherited, truncating MAGEL2 mutations.

While I think that your study merits publication, I do not concur with your statement that

“Our genome-edited mouse model could serve as a new model of truncated Magel2 to investigate the mechanism of SYS,” as there are no specific indications how the mechanisms in SYS should be addressed.

I would like you to either produce a specific list of experiments to be conducted as the next steps, in your genome-edited model mouse to address the mechanisms underlying SYS,

OR to remove the above statement from the manuscript.

I am not asking you to conduct further experiments.

Please also take the comments of the Reviewers into account, when submitting a revised version of the manuscript.

We would appreciate receiving your revised manuscript by Jul 03 2020 11:59PM. To enhance the reproducibility of your results, we recommend that if applicable you deposit your laboratory protocols in protocols.io, where a protocol can be assigned its own identifier (DOI) such that it can be cited independently in the future. For instructions see: http://journals.plos.org/plosone/s/submission-guidelines#loc-laboratory-protocols

We look forward to receiving your revised manuscript.

Kind regards,

Andreas R. Janecke, M.D.

Academic Editor

PLOS ONE

Journal Requirements:

2. To comply with PLOS ONE submissions requirements, in your Methods section, please provide additional information on the animal research and ensure you have included details on (1) methods of sacrifice, (2) methods of anesthesia and/or analgesia, and (3) efforts to alleviate suffering."

3. In your Methods section, please include a comment about the state of the animals following this research. Were they euthanized or housed for use in further research? If any animals were sacrificed by the authors, please include the method of euthanasia and describe any efforts that were undertaken to reduce animal suffering.

5. Thank you for stating in your Funding Statement:

"This study was partly supported by JSPS KAKENHI Grant-in-Aid for Early-Career

Scientists (JP18K15682) awarded to YN. The funders had no role in study design, data

collection and analysis, decision to publish, or preparation of the manuscript."

Reviewers' comments:

Reviewer's Responses to Questions

**Comments to the Author**

1. Is the manuscript technically sound, and do the data support the conclusions?

Reviewer #1: Partly

Reviewer #2: Yes

2. Has the statistical analysis been performed appropriately and rigorously? 

Reviewer #1: N/A

Reviewer #2: Yes

3. Have the authors made all data underlying the findings in their manuscript fully available?

Reviewer #1: Yes

Reviewer #2: Yes

4. Is the manuscript presented in an intelligible fashion and written in standard English?

Reviewer #1: Yes

Reviewer #2: Yes

5. Review Comments to the Author

Reviewer #1: Truncating variants in the paternal allele of MAGEL2 have previously been reported, and described as resulting in Schaaf-Yang syndrome (SYS), The phenotypes of SYS patients overlap those of Prader-Willi syndrome patients, which include neonatal hypotonia, feeding problems and mental retardation. Ieda et al report the generation and phenotypes of two mouse models for SYS, one representing a CRISPR-Cas generated frameshift mutation within Magel2, the second representing an expressed truncated version of Magel2 thus mimicking a frameshift mutation in the Magel2 gene which reflects common SYS mutations in humans. Since the phenotype of Magel2 null mutations in humans is milder than those of frameshift mutations (reflecting many aspects of PWS) the authors hypothesized that truncated versions of Magel2 might exert a dominant negative gain-of-function in humans and thus might be the reason for more severe SYS phenotypes. While the Magel2 frameshift mice almost reflected wt mice the – ubiquitously – expressed truncated version of Magel2 resulted in fetal or neonatal death.

Major Concerns:

1)The authors expressed the truncated version of Magel2 ubiquitously in all tissues , although Magel2 is normally expressed mainly in brain. Due to the small number of transgenic truncated specimens (n=3) the authors were unable to assess the patho-mechanism of the fetal or neonatal death, being due to either overexpression of the truncated protein or ubiquitous expression; i.e. this is a “no data experiment” since no conclusions can be drawn. The authors are strongly advised to closely investigate the patho-mechanism of fetal or neonatal death of the truncated version of Magel2, since this was the major objective in their study.

2) The CRIPR-Cas generated frameshift Magel2 mice, after four weeks, did not show obvious abnormalities in physical findings and failed to recapitulate the common phenotype of SYS. Their value in investigating pathophysiology in SYS is thus extremely limited. Other models for the effects of the more severe phenotype of Magel2 frameshift mutations should ,be considered and discussed.

Minor Concers:

The authors should cite relevant primary literature regarding PWS etiologies; it is common practice in recent years to cite reviews which again cite reviews and so on, but not mention the original publications: for example, the group of Hüttenhofer (Cavaille et al, 2000 PNAS) discovered the ncRNAs (SNORD115, 116) later shown by the group of Beaudet (Nature Gen. 2008) to be solely responsible for PWS. Please cite relevant original literature , not reviews.

Reviewer #2: This study reports the generation of two new mouse models for Schaaf-Yang syndrome: an over expression model for an N-Terminal truncated Magel2 protein, and a transgenic mouse with a truncation mutation. These models may be important for the investigation of SYS in a preclinical model. However, the models presented have only undergone limited characterization.

Major comments:

I did not see a quantification of the level of overexpression for the overexpression mouse model.

The decreased number of life-born pups with the frameshifting variant of Magel2 on the paternal allele may indicate prenatal lethality, as has been suggested for the Magel2-LacZ mouse . A power calculation should be done and subsequently that hypothesis should be tested.

Minor comment:

Lines 87 and 390: MAGEL2 is misspelled.

6. PLOS authors have the option to publish the peer review history of their article (what does this mean?). If published, this will include your full peer review and any attached files.

Reviewer #1: No

Reviewer #2: No

---

## [Author Response · Author response to Decision Letter 0]

6 Jul 2020

Thank you for inviting us to submit a revised manuscript entitled, ‘Two mouse models carrying truncating mutations in Magel2 show distinct phenotypes.’ to PLOS ONE. We have incorporated changes reflecting the suggestions you have graciously provided. We also hope that our edits and the responses we provide below satisfactorily address the concerns raised by you and the reviewers.

We hope that you find the revised manuscript acceptable.

---

## [Editor Report · Decision Letter 1]

4 Aug 2020

Two mouse models carrying truncating mutations in Magel2 show distinct phenotypes

PONE-D-20-10964R1

Dear Dr. Saitoh,

We’re pleased to inform you that your manuscript has been judged scientifically suitable for publication and will be formally accepted for publication once it meets all outstanding technical requirements.

Kind regards,

Andreas R. Janecke, M.D.

Academic Editor

PLOS ONE
---

## [Editor Report · Acceptance letter]

7 Aug 2020

PONE-D-20-10964R1 

Two mouse models carrying truncating mutations in Magel2 show distinct phenotypes 

Dear Dr. Saitoh:

I'm pleased to inform you that your manuscript has been deemed suitable for publication in PLOS ONE. Congratulations! Your manuscript is now with our production department. 

Kind regards, 

on behalf of

Dr. Andreas R. Janecke 

Academic Editor

PLOS ONE